# The effect of a telephone-based intervention on physical activity after stroke

**Seungwoo Cha**[1,2], **Won Kee Chang**[1,2], **Hee-Mun Cho**[1,2], **Yun-Sun Jung**[1], **Miji Kang**[1,2], **Nam-Jong Paik**[1,2], **Won-Seok Kim**[1,2]*

**1** Department of Rehabilitation Medicine, Seoul National University College of Medicine, Seoul National University Bundang Hospital, Seongnam, Korea, **2** Gyeonggi Regional Cardiocerebrovascular Disease Center, Seoul National University Bundang Hospital, Seongnam, Korea

* wondol77@gmail.com

## Abstract

Despite the effectiveness of telephone-based interventions for medical adherence and improved blood pressure, studies on the effect of such interventions on physical activity (PA) are needed. Therefore, we investigated the impact of a telephone-based intervention on PA in patients with subacute stroke. This pre-post study included patients who participated in an education program for stroke rehabilitation before being discharged to home, with a modified Rankin scale (mRS) score of ≤ 3. Patients hospitalized in 2020 (intervention group) received a nurse-led telephone-based intervention with a PA measurement once monthly during the 3 months after discharge. Those hospitalized in 2019 (historical controls) only received a PA measurement 3 months after discharge. Physical activity was assessed via a questionnaire by phone. In addition, demographics, medical history, smoking, mRS scores, and Patient Health Questionnaire-9 data were collected. The study included 139 participants (73 in intervention, 66 in control). The intervention group had a higher proportion of patients with mRS of 0–1 and a shorter length of hospital stay than the historical controls. Three months post-discharge, a significantly higher proportion of participants were physically active in the intervention group (48 [71.6%] vs. control group, 25 [34.7%]). In addition, the intervention group had a significantly higher median energy expenditure (924 vs. 297 MET-min/week) than the control group. The OR of the intervention for achieving 'physically active' individuals was 4.749 (95% CI, 2.313–9.752) before and 5.222 (95% CI, 1.892–14.419) after adjusting for possible confounders. A telephone-based intervention improved PA three months after stroke. Further studies with larger sample size and long-term follow-up are needed.

## Introduction

The level of physical activity (PA) is low regardless of the time following stroke [1], and stroke survivors report even lower PA than elderly patients with chronic diseases [2]. However, PA benefits health-related outcomes after stroke and cardiovascular risk factor management [3, 4]. PA also supports social participation, helping stroke survivors to achieve their goals and

**Data Availability Statement:** The data used for this study could not be made publicly available due to legal constraints, specifically the Personal Information Protect Act (2014). The pseudonymized data that support the findings of

this study are available from the corresponding author, Pf. Won-Seok Kim, or the IRB of Seoul National University Bundang Hospital (snubhirb@gmail.com) upon reasonable request, subsequent approval from the local IRB, and completion of a legal data sharing agreement.

**Funding:** The author(s) received no specific funding for this work.

**Competing interests:** The authors have declared that no competing interests exist.

adjust to life after stroke [5]. Therefore, current guidelines for secondary stroke prevention recommend regular PA [6].

Previous studies have revealed the feasibility and effectiveness of telephone-based interventions in patients with stroke. A randomized controlled study reported that goal-setting telephone follow-up improved medication adherence in ischemic stroke [7]. Another study revealed that nurse-led, telephone-based intervention was efficient in improving blood pressure and low-density lipoprotein levels after stroke [8]. However, telephone-based interventions are not routinely performed as a part of stroke rehabilitation, and studies on the effect of the telephone-based intervention on post-stroke PA are needed.

Therefore, this pre-post study using historical controls aimed to investigate the effects of telephone-based interventions on PA in patients with subacute stroke.

## Materials and methods

### Study population

The study screened patients who were admitted to a tertiary hospital in Korea (Seoul National University Bundang Hospital) for acute stroke between 2019 and 2020 and participated in an education program for stroke rehabilitation (general information on the pathophysiology, prognosis, and secondary prevention of stroke, and diet recommendations, and PA were provided in this program). Those with a modified Rankin Scale (mRS) score of 0–3 (from no symptoms to moderate disability) [9] and were discharged to home were included in the study. When patients were hospitalized for rehabilitation after staying at home, did not respond to a telephone call, or refused to participate in the study, they were excluded (S1 Fig).

Data were anonymized for analysis; thus, the need for informed consent was waived. The study was reviewed by the institutional review board of Seoul National University Bundang Hospital (IRB number B-2206-762-111).

### Intervention

Patients hospitalized in 2020 received an education program before discharge and a telephone-based intervention once monthly for 3 months after discharge by a trained nurse (Fig 1). The educational program consisted of consultation and explanation regarding the general benefits of rehabilitation. In addition, information about recovery stages, functional status, and precautions or recommendations on daily activities were provided.

During the intervention, education and discussions were implemented to encourage post-stroke PA. PA was monitored through a phone interview (at discharge and 1, 2, and 3 months after discharge), and the target PA level was determined after discussion with the patients. Pain, diet, sleep, and other barriers to PA were identified and adjusted. When weakness, dizziness, and other symptoms related to stroke recurrence were observed, a neurology consultation was offered immediately. The intervention lasted for at least 15 min. Examples of frequently used questions in the intervention are presented in S1 Table.

However, patients hospitalized in 2019 received an education program before discharge and were monitored for PA only once at 3 months after discharge (Fig 1).

### Physical activity

Participants responded to a questionnaire regarding PA via telephone (four times for patients hospitalized in 2020 and once for patients hospitalized in 2019). The Korean version of the International Physical Activity Questionnaire (IPAQ) was used [10]. PA-related energy expenditure (MET-min/week) was calculated using PA intensity, frequency, and duration.

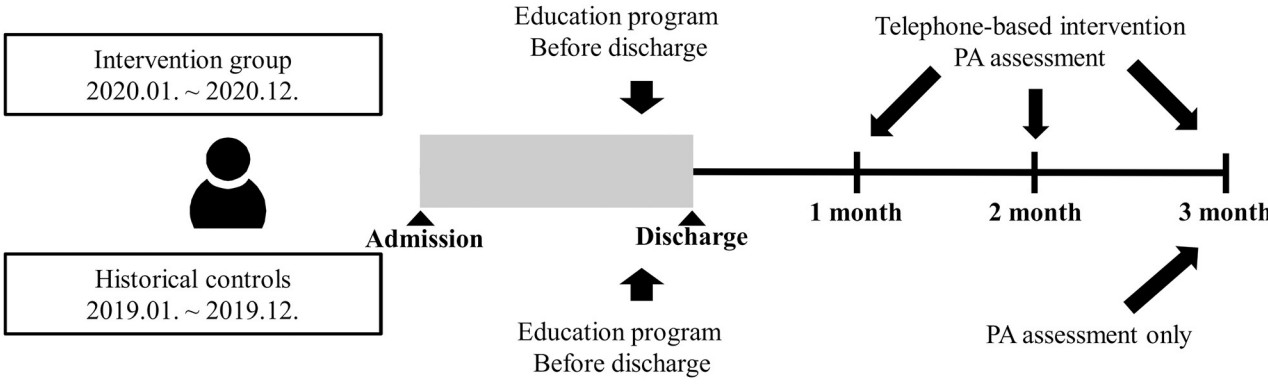

**Fig 1. Study flowchart.**

Participants were considered physically active if they engaged in health-enhancing PA or were minimally active (S2 Table). The change in PA level was classified into improved, stationary, and aggravated in the intervention group.

## Other covariates

Sociodemographic data on age, sex, body mass index (BMI), education, marital status, occupation, and medical insurance were collected. After reviewing the imaging findings, stroke type and cortical involvement were determined. The presence of aphasia was defined as 1 or more on the National Health Institute Stroke Scale item 9 for best language (range 0–3) during admission. In addition, a history of hypertension, diabetes mellitus, atrial fibrillation, and stroke were investigated. Participants currently smoking were defined as current smokers; otherwise, they were defined as nonsmokers. The Patient Health Questionnaire-9 (PHQ-9) scores were evaluated at the time of discharge and the mRS score was measured at discharge and 3 months after discharge. Changes in the mRS score between discharge and 3 months were classified into improved, stationary, and aggravated for analysis.

## Statistics

Baseline characteristics are reported as means with standard deviations or as numbers with percentages. The Student's t-test and the Chi-square test analyzed continuous (after identifying a normal distribution) and categorical variables, respectively. PA-related energy expenditure was compared between the intervention and control groups using the Wilcoxon rank-sum test because the variable was not normally distributed. Logistic regression analysis examined the effect of the intervention on PA. Effect sizes are expressed as ORs with 95% CIs. The results were adjusted for age and sex (adjusted ORs) or age, sex, stroke type, cortical involvement, aphasia, a history of hypertension, diabetes mellitus, atrial fibrillation, and stroke, BMI, smoking, education level, marital status, occupation, mRS and PHQ-9 at discharge, and length of stay (fully adjusted ORs). Two-sided $p$ values of $< .05$ were considered significant. The statistical analyses were performed using IBM SPSS Statistics version 25.0 (Armonk, NY, USA).

## Results

Seventy-three patients received the intervention and were included in the pre-post analysis, while 66 were included as historical controls (S1 Fig). The intervention group had a significantly higher proportion of patients with an mRS score of 0–1 at discharge and a shorter length of stay than the control group (Table 1). However, there was no significant difference between

**Table 1. Comparison of the baseline characteristics.**

| | Intervention (n = 73) | Control (n = 66) | *p*-value |
|---|---|---|---|
| **Age, years** | 63.7 ± 12.8 | 65.1 ± 13.0 | .51 |
| **Male, n (%)** | 53 (72.6%) | 43 (65.2%) | .34 |
| **Stroke type, n (%)** | | | .40 |
| Ischemic | 63 (86.3%) | 60 (90.9%) | |
| Hemorrhagic | 10 (13.7%) | 6 (9.1%) | |
| **Cortical involvement, n (%)** | | | .62 |
| Yes | 29 (39.7%) | 29 (43.9%) | |
| No | 44 (60.3%) | 37 (56.1%) | |
| **Aphasia, n (%)** | | | .78 |
| Yes | 10 (13.7%) | 8 (12.1%) | |
| No | 63 (86.3%) | 58 (87.9%) | |
| **Past medical history, n (%)** | | | |
| Hypertension | 57 (78.1%) | 48 (72.7%) | .46 |
| Diabetes mellitus | 20 (27.4%) | 22 (33.3%) | .45 |
| Atrial fibrillation | 10 (13.7%) | 7 (10.6%) | .58 |
| Previous stroke | 12 (16.4%) | 12 (18.2%) | .79 |
| **Body mass index, n (%)** | | | .18 |
| < 18.5 kg/m$^2$ | 1 (1.4%) | 4 (6.1%) | |
| 18.5–24.9 kg/m$^2$ | 42 (57.5%) | 42 (63.6%) | |
| ≥ 25 kg/m$^2$ | 30 (41.1%) | 20 (30.3%) | |
| **Smoking, n (%)** | | | .42 |
| Current smoker | 21 (28.8%) | 15 (22.7%) | |
| Non-smoker | 52 (71.2%) | 51 (77.3%) | |
| **Education level, n (%)** | | | .45 |
| < Elementary school | 12 (16.4%) | 9 (13.6%) | |
| Middle school | 14 (19.2%) | 7 (10.6%) | |
| High school | 17 (23.3%) | 20 (30.3%) | |
| > College | 30 (41.1%) | 30 (45.5%) | |
| **Marital status** | | | .33 |
| Married | 65 (89.0%) | 55 (83.3%) | |
| Others | 8 (11.0%) | 11 (16.7%) | |
| **Occupation** | | | .53 |
| Yes | 36 (49.3%) | 29 (43.9%) | |
| Others | 37 (50.7%) | 37 (56.1%) | |
| **mRS at discharge** | | | .003* |
| 0–1 | 19 (26.0%) | 4 (6.1%) | |
| 2 | 20 (27.4%) | 16 (24.2%) | |
| 3 | 34 (46.6%) | 46 (69.7%) | |
| **PHQ-9 at discharge** | | | .38 |
| 0–4 (No depression) | 35 (51.5%) | 37 (63.8%) | |
| 5–9 (Mild) | 19 (27.9%) | 12 (20.7%) | |
| > 10 (Moderate to severe) | 14 (20.6%) | 9 (15.5%) | |
| **Length of stay, days** | 16.5 ± 9.0 | 23.3 ± 10.7 | < .001 |

mRS, modified Rankin Scale; PHQ-9, Patient Health Questionnaire-9.

*$p < .05$.

both groups in other parameters. Aphasia was present in 18 patients (12.9%); most were mild non-fluent or anomic type.

Three months after discharge, a significantly higher proportion of participants in the intervention group (n = 48, 71.6%) were physically active compared to those in the control group (n = 25, 34.7%). The median energy expenditure was significantly higher in the intervention group (924 MET-min/week) than in the control group (297 MET-min/week) (Fig 2).

A notable proportion of patients in the intervention group changed their PA level during the follow-up. One month after discharge, 28 (65.1%) previously inactive participants became physically active, while 6 (20.0%) participants became inactive. Few participants changed their PA level 2 or 3 months after discharge. Changes in PA level between consecutive time points are illustrated in Fig 3.

The OR of the intervention for improving PA at 3 months was 4.749 (95% CI, 2.313–9.752) in the crude model and 5.222 (95% CI, 1.892–14.419) in the fully adjusted model. In the subgroup analysis, patients with an mRS score of 2–3 at discharge had an OR of 4.518 (95% CI,

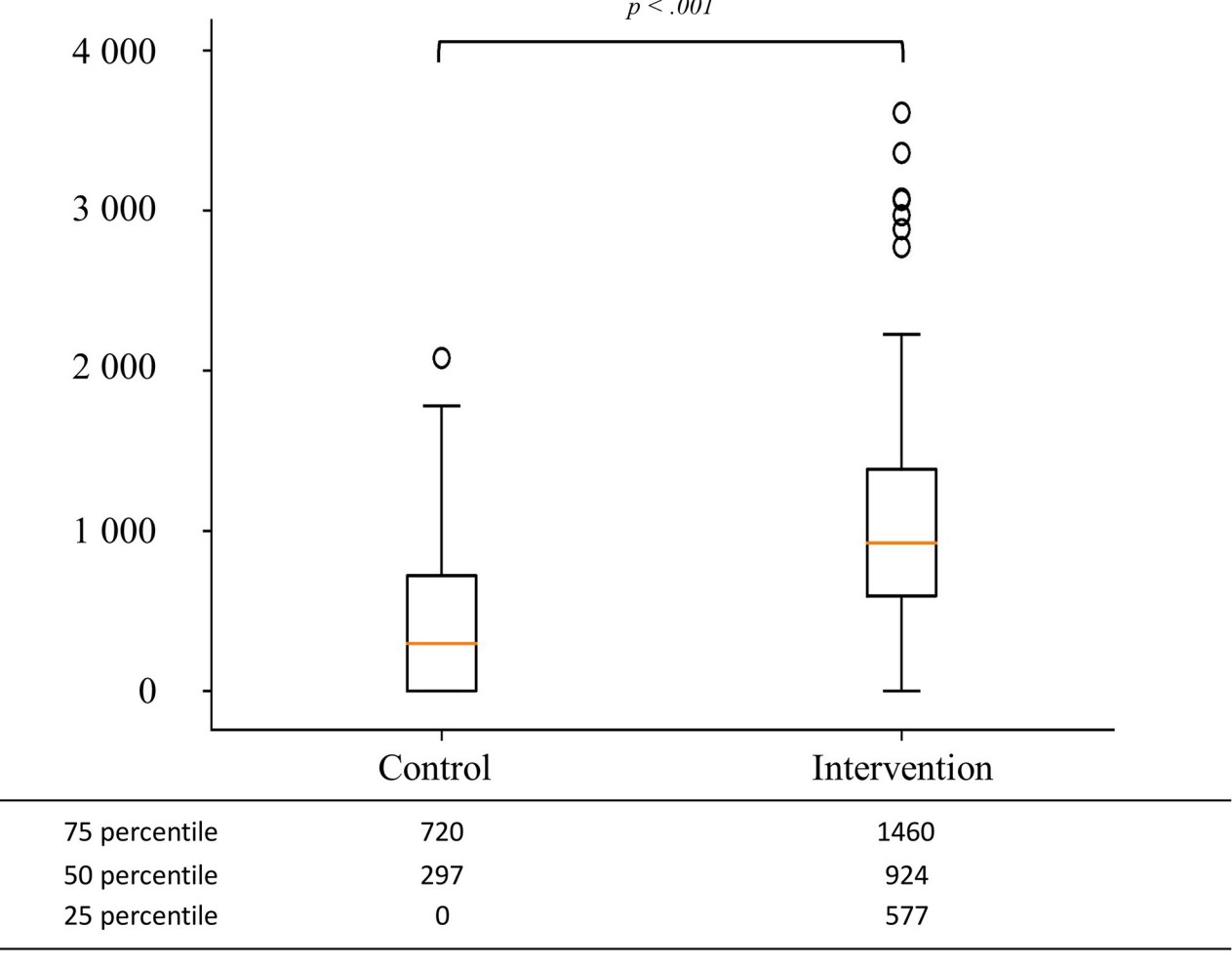

| | Control | Intervention |
|---|---|---|
| 75 percentile | 720 | 1460 |
| 50 percentile | 297 | 924 |
| 25 percentile | 0 | 577 |

**Fig 2. Physical activity-related energy expenditure in the intervention and control groups.**

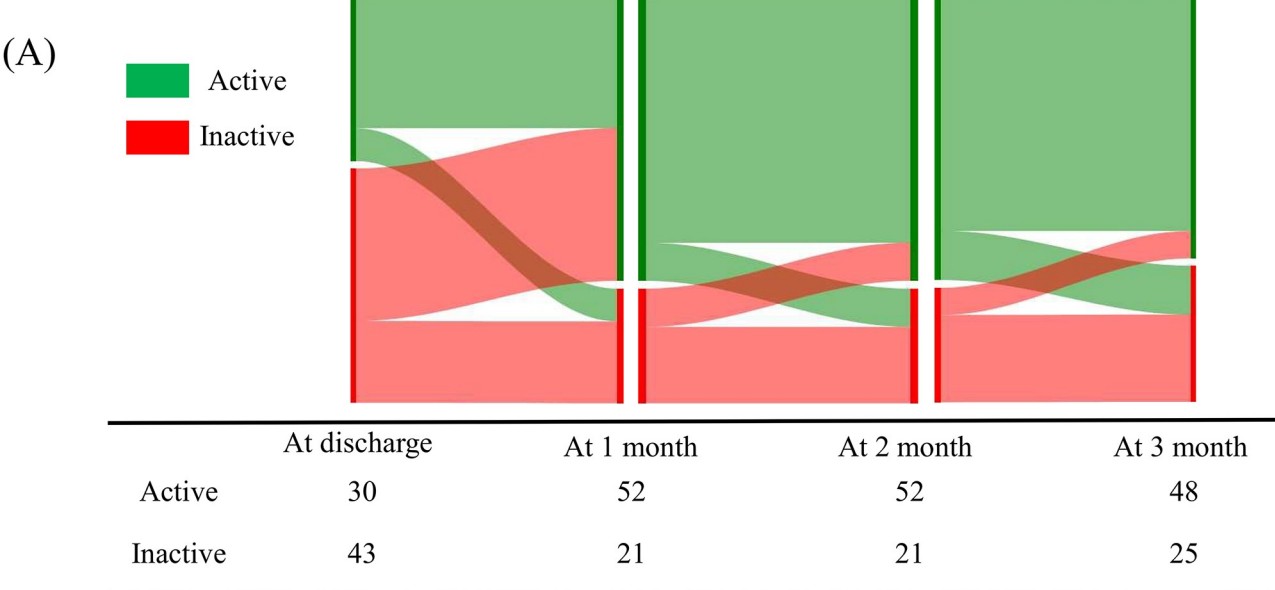

| | At discharge | At 1 month | At 2 month | At 3 month |
|---|---|---|---|---|
| Active | 30 | 52 | 52 | 48 |
| Inactive | 43 | 21 | 21 | 25 |

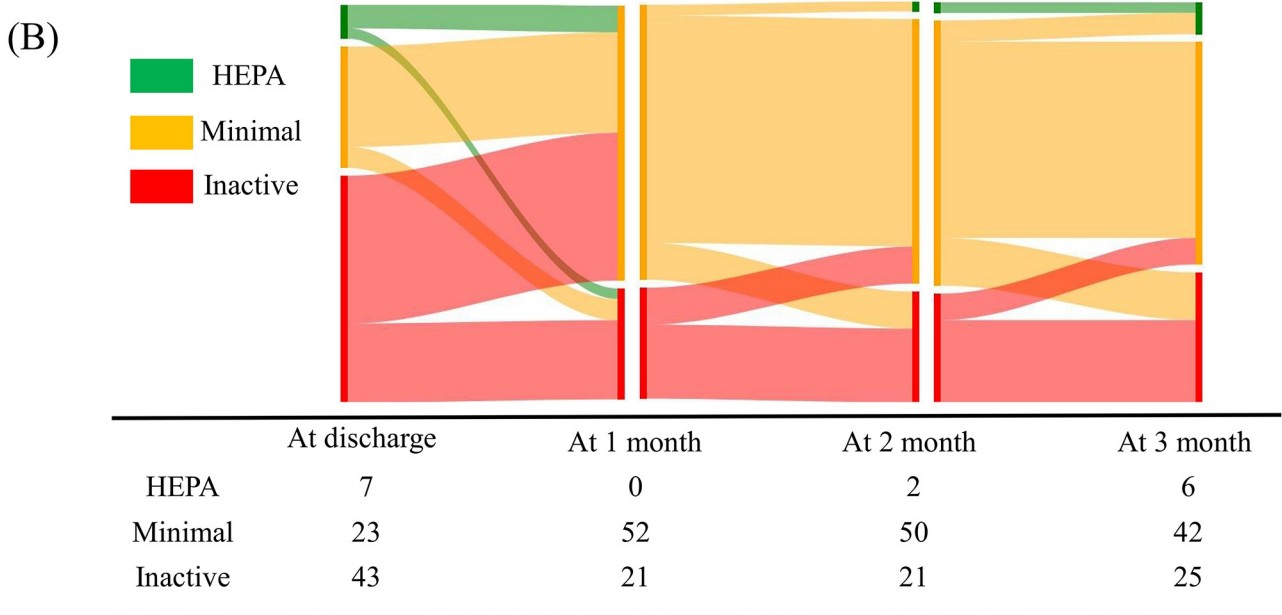

| | At discharge | At 1 month | At 2 month | At 3 month |
|---|---|---|---|---|
| HEPA | 7 | 0 | 2 | 6 |
| Minimal | 23 | 52 | 50 | 42 |
| Inactive | 43 | 21 | 21 | 25 |

**Fig 3.** Changes in physical activity ((A): Active and Inactive, (B): Health-enhancing physical activity (HEPA), Minimally active, and Inactive) in the intervention group from discharge to follow-up at 3 months.

2.052–9.947) in the crude model and 5.150 (95% CI, 1.758–15.085) in a fully adjusted model. Patients with an mRS score of 3 had a higher OR of 7.636 (95% CI 2.805–20.791) in the crude model and 22.868 (3.611–144.798) in the fully adjusted model (Table 2).

Patients who were active 3 months after discharge were more likely to have received the intervention and had a significantly higher proportion of participants with an mRS score of 0–1 at discharge than those who were inactive. In addition, BMI was higher, and length of stay was shorter in those who were active (S3 Table).

**Table 2. The odds ratio of intervention for physical activity at 3 months.**

|  | Group | n | Crude OR (95% CI) | *p*-value | Adjusted OR† (95% CI) | *p*-value | Fully adjusted OR‡ (95% CI) | *p*-value |
|---|---|---|---|---|---|---|---|---|
| mRS 0–3 (n = 139) | Intervention | 73 | 4.749 (2.313–9.752) | < .001 | 4.676 (2.272–9.624) | < .001 | 5.143 (1.837–14.400) | .002 |
|  | Control | 66 | Reference | | | | | |
| mRS 2–3 alone (n = 116) | Intervention | 54 | 4.518 (2.052–9.947) | < .001 | 4.547 (2.059–10.042) | < .001 | 5.643 (1.823–17.467) | .003 |
|  | Control | 62 | Reference | | | | | |
| mRS 3 alone (n = 80) | Intervention | 34 | 7.636 (2.805–20.791) | < .001 | 7.213 (2.625–19.818) | < .001 | 23.344 (3.320–164.143) | .002 |
|  | Control | 46 | Reference | | | | | |

mRS, modified Rankin Scale.

†Adjusted for age and sex.

‡Adjusted for age, sex, stroke type, cortical involvement, aphasia, a history of hypertension, diabetes mellitus, atrial fibrillation, and stroke, body mass index, smoking, education level, marital status, occupation, modified Rankin Scale, Patient Health Questionnaire-9, and length of stay.

When the change in mRS score between discharge and 3 months was divided into improved, stationary, and aggravated, no significant difference was observed between the intervention and control groups. In the intervention group, the change in PA did not significantly differ based on changes in mRS scores (S4 Table).

The comparison of demographics between the intervention group and excluded participants during the intervention period in 2020 (due to readmission after discharge, no response, and refusal to participate, n = 32) revealed that the excluded participants had more hemorrhagic-type stroke, lower BMI, and a lower proportion of married patients. However, other factors, such as age, sex, mRS score, PHQ-9 score, and length of stay, did not significantly differ between both groups (S5 Table).

## Discussion

This study revealed that a telephone-based intervention was applicable in clinical settings and effectively improved for PA 3 months after discharge in patients who suffered an acute stroke. Participants who participated in the intervention were approximately five times more likely to be physically active, after adjusting for possible confounders. Specifically, the intervention group had a higher proportion of patients with an mRS score of 0–1 (no symptoms or slight disability) and a shorter length of stay than the control group. However, other demographic factors did not differ, implying that disease severity may vary between both groups. However, the subgroup analysis excluding those with mRS scores of 0–1 revealed consistent results. In addition, changes in the level of PA did not significantly differ based on changes in mRS scores of the intervention group, suggesting that an increase in PA would result from the intervention rather than improved mRS scores during stroke recovery.

Telephone delivery is a cost-effective and easily accessible method that provides repeated contact to promote behavioral changes [11]. Hence, a telephone-delivered intervention was attempted and successfully improved PA and dietary habits in various populations, such as cancer survivors or sedentary older adults [12]. For patients who suffered a stroke, a nurse-led telephone-based intervention once monthly improved blood pressure and low-density lipoprotein C levels 36 months after discharge [13]. It also resulted in a fluctuating but gradual increase in participants who reached the target levels during the intervention. Our study revealed that approximately two-thirds of physically inactive patients in the intervention group became active after a month. In addition, some patients increased their level of PA 2 or 3 months after discharge, although the overall proportion decreased (Fig 3). Therefore, persistent intervention may be needed to maintain PA and its long-term benefits.

Before prescribing the intervention, it is essential to address the barriers to post-stroke PA. Commonly reported barriers include transportation, access, cost, stroke-related impairment, embarrassment, fear of recurrent stroke, disabilities, and lack of knowledge [14]. Advance age, fatigue, depression, and poor health-related quality of life have also been associated with low post-stroke PA [15]. In addition, a study regarding sex disparities in post-stroke PA revealed that PA level is lower in women than in men [16]. In our study, higher mRS scores at discharge, low BMI, and longer length of stay might be barriers to PA 3 months after discharge. Therefore, prescribing individually tailored education about PA after identifying associated barriers would help improve efficacy and adherence to exercise interventions. A telephone-based intervention can be delivered to remote areas, costs less, and is not constrained by space; therefore, it can help eliminate some barriers.

This study had a few limitations. First, selection bias may have been present because only patients hospitalized in 2020 were offered an intervention. The length of stay and mRS scores in the intervention and control groups differed, although other baseline characteristics did not. This is partially because more patients with mild symptoms were recently transferred to the rehabilitation department for early stroke rehabilitation. However, the overall clinical setting for stroke rehabilitation did not change between 2019 and 2020. In addition, our results were robust after adjusting for possible confounders such as age, sex, socioeconomic status, and past medical history. Subgroup regression analysis was additionally performed, showing a consistent result, to exclude the effect of very mild participants whose mRS score was 0 or 1. Second, PA was assessed through an interview using the IPAQ. PA level may have been overestimated; however, the Korean version of the IPAQ has been validated based on accelerometer measurements [17]. Third, there was no blinding regarding the IPAQ assessments. However, the IPAQ is a tool based on direct responses from a patient and was objectively assessed by the nurse using the questions in the questionnaire. Finally, long-term follow-up data were not collected, although repetitive and persistent encouragement may be required to improve and maintain PA. As a telephone-based intervention is not expensive and easy to perform in clinical settings, further studies with longer follow-ups could be performed.

## Conclusion

A nurse-led telephone-based intervention to promote PA after discharge in patients with subacute stroke was feasible and associated with increased PA 3 months after discharge. However, future studies with larger sample sizes and long-term follow-up with a parallel design are needed.

## Supporting information

**S1 Fig. Patient flow.**
(TIF)

**S1 Table. Questions frequently used in the intervention.**
(DOCX)

**S2 Table. Physical activity categorization.**
(DOCX)

**S3 Table. Comparison of baseline characteristics between the active and inactive group three months after discharge.**
(DOCX)

**S4 Table. The change in mRS scores according to the group (intervention vs. control) or the change in PA.**
(DOCX)

**S5 Table. Comparison between the intervention group and excluded participants (readmission, non-responder, and disagreement) admitted in 2020.**
(DOCX)

## Author Contributions

**Conceptualization:** Seungwoo Cha, Nam-Jong Paik.

**Data curation:** Seungwoo Cha, Won Kee Chang, Hee-Mun Cho, Yun-Sun Jung, Miji Kang, Won-Seok Kim.

**Formal analysis:** Seungwoo Cha, Won Kee Chang, Yun-Sun Jung, Miji Kang, Won-Seok Kim.

**Investigation:** Seungwoo Cha, Won Kee Chang, Nam-Jong Paik, Won-Seok Kim.

**Methodology:** Seungwoo Cha, Won Kee Chang, Hee-Mun Cho, Yun-Sun Jung, Miji Kang, Nam-Jong Paik, Won-Seok Kim.

**Project administration:** Nam-Jong Paik, Won-Seok Kim.

**Resources:** Seungwoo Cha, Hee-Mun Cho, Yun-Sun Jung, Miji Kang.

**Software:** Seungwoo Cha, Miji Kang.

**Supervision:** Seungwoo Cha, Won Kee Chang, Nam-Jong Paik, Won-Seok Kim.

**Validation:** Seungwoo Cha, Hee-Mun Cho, Miji Kang, Won-Seok Kim.

**Visualization:** Seungwoo Cha, Hee-Mun Cho, Yun-Sun Jung.

**Writing – original draft:** Seungwoo Cha, Hee-Mun Cho, Won-Seok Kim.

**Writing – review & editing:** Seungwoo Cha, Won Kee Chang, Hee-Mun Cho, Yun-Sun Jung, Miji Kang, Nam-Jong Paik, Won-Seok Kim.

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
