## [Decision Letter · Decision Letter 0]

14 Sep 2022

PONE-D-22-17677The Effect of a Telephone-based Intervention on Physical Activity after StrokePLOS ONE

Dear Dr. Kim,

Thank you for submitting your manuscript to PLOS ONE. After careful consideration, we feel that it has merit but does not fully meet PLOS ONE’s publication criteria as it currently stands. Therefore, we invite you to submit a revised version of the manuscript that addresses the points raised during the review process.

We look forward to receiving your revised manuscript.

Kind regards,

Billy Morara Tsima, MD MSc

Academic Editor

PLOS ONE

Journal Requirements:

Reviewers' comments:

Reviewer's Responses to Questions

**Comments to the Author**

1. Is the manuscript technically sound, and do the data support the conclusions?

Reviewer #1: Yes

Reviewer #2: Yes

2. Has the statistical analysis been performed appropriately and rigorously? 

Reviewer #1: Yes

Reviewer #2: Yes

3. Have the authors made all data underlying the findings in their manuscript fully available?

Reviewer #1: Yes

Reviewer #2: Yes

4. Is the manuscript presented in an intelligible fashion and written in standard English?

Reviewer #1: Yes

Reviewer #2: Yes

5. Review Comments to the Author

Reviewer #1: １．Depression is evaluated by PHQ-9, but evaluation of comprehension and memory ability for telephone intervention may affect the results. Therefore, it is desirable to add the location of the lesion (cortical or subcortical) and the presence or absence of aphasia (especially the presence or absence of comprehension disorders) to the comparison of baseline characteristics.

２．It is unclear whether the band of the curve in Fig.3 shows a continuous change or a discontinuous change. The curve is shown as active to inactive after 1 month, active again after 2 months, and inactive again after 3 months. Is that interpretation correct?

３．It would be better to evaluate and analyze changes in the degree of PA (Health enhancing PA/Minimally active).

４．Although the purpose of this study is the effect of a telephone-based intervention on PA, it is desirable to consider the relationship between PA improvement and mRS improvement in the discussion.

Reviewer #2: The authors present an important topic of nurse intervention and telephone monitoring of the health and physical activity of patients who have recently had a stroke. The authors rightly noted that physical activity provides benefits for health-related outcomes after stroke and cardiovascular risk factor management. Physical activity also supports social participation, helping stroke survivors to achieve their goals and adjust to life after stroke.

Of course, telephone interventions limit the amount of information that can be gathered about a patient's current condition, but how previous studies have shown the feasibility and effectiveness of telephone-based interventions in patients after stroke. The study is designed well. The title and abstract cover the main aspect of the work. The introduction provides background and information relevant to the study. The methodology are described very exactly and clearly. The statistical analysis is performed correctly. The results are described accurately and clearly. The authors accurately described the limitations of the study.

The manuscript is written correctly. The authors rightly conclude that further studies with a larger number of patients and long-term follow-up are needed. The weak points of the study are that there was the single-center study, hospital-based retrospective study and the sample size was small.

6. PLOS authors have the option to publish the peer review history of their article (what does this mean?). If published, this will include your full peer review and any attached files.

Reviewer #1: No

Reviewer #2: **Yes: **Piotr Sobolewski

---

## [Author Response · Author response to Decision Letter 0]

27 Sep 2022

#Comment 1-1 

Depression is evaluated by PHQ-9, but evaluation of comprehension and memory ability for telephone intervention may affect the results. Therefore, it is desirable to add the location of the lesion (cortical or subcortical) and the presence or absence of aphasia (especially the presence or absence of comprehension disorders) to the comparison of baseline characteristics.

Response: We thank the reviewer for this comment. Cortical involvement was determined after reviewing imaging findings. Aphasia was defined based on the NIHSS item 9 for best language during admission. In the fully adjusted model for the odds ratio of the intervention, the cortical involvement and aphasia were also adjusted. 

#Comment 1-2 

It is unclear whether the band of the curve in Fig.3 shows a continuous change or a discontinuous change. The curve is shown as active to inactive after 1 month, active again after 2 months, and inactive again after 3 months. Is that interpretation correct?

Response: We thank the reviewer for this comment. The level of physical activity was measured four times (at discharge and 1, 2, and 3 months after discharge). Fig. 3 (Sankey diagram) illustrates the changes in physical activity level between consecutive time points, which are discontinuous. Therefore, Fig. 3 was revised to reveal the discontinuity clearly, and the sentences about physical activity evaluation have been added. 

#Comment 1-3

It would be better to evaluate and analyze changes in the degree of PA (Health enhancing PA/Minimally active).

Response: Thank you for your comment. Few patients displayed health-enhancing PA as illustrated in the revised Fig 3 (B) (Green = Health-enhancing PA, Orange = Minimally active, and Red = Inactive). The number of patients with health-enhancing PA was 7, 0, 2, and 6 at discharge, 1, 2, and 3 months after discharge, respectively. Therefore, it was challenging to analyze the clinical characteristics of those with health-enhancing PA separately. Instead, we revised Fig 3 to include changes in PA degree ((A): active and inactive, (B): health-enhancing physical activity, minimally active, and inactive).

#Comment 1-4 

Although the purpose of this study is the effect of a telephone-based intervention on PA, it is desirable to consider the relationship between PA improvement and mRS improvement in the discussion.

Response: Thank you for your comment. Baseline mRS scores differed between the intervention and control groups; however, mRS score improvement within 3 months after discharge was not significantly different between both groups. In addition, mRS score improvement did not significantly differ based on PA changes in the intervention group (the analysis was conducted only in the intervention group because the PA level at discharge was not evaluated in the control group). This may suggest that PA improvement results from the intervention rather than PA improvement during stroke recovery. The table is presented as S4 Table.

#Comment 2-1

The authors present an important topic of nurse intervention and telephone monitoring of the health and physical activity of patients who have recently had a stroke. The authors rightly noted that physical activity provides benefits for health-related outcomes after stroke and cardiovascular risk factor management. Physical activity also supports social participation, helping stroke survivors to achieve their goals and adjust to life after stroke.

Of course, telephone interventions limit the amount of information that can be gathered about a patient's current condition, but how previous studies have shown the feasibility and effectiveness of telephone-based interventions in patients after stroke. The study is designed well. The title and abstract cover the main aspect of the work. The introduction provides background and information relevant to the study. The methodology are described very exactly and clearly. The statistical analysis is performed correctly. The results are described accurately and clearly. The authors accurately described the limitations of the study.

Response: We thank the reviewer for this comment. We agree with the comment and believe that our study, consistent with previous studies, supports the telephone-based intervention on physical activity improvement in patients with stroke. Thank you.

---

## [Editor Report · Decision Letter 1]

5 Oct 2022

The Effect of a Telephone-based Intervention on Physical Activity after Stroke

PONE-D-22-17677R1

Dear Dr. Kim,

We’re pleased to inform you that your manuscript has been judged scientifically suitable for publication and will be formally accepted for publication once it meets all outstanding technical requirements.

Kind regards,

Billy Morara Tsima, MD MSc

Academic Editor

PLOS ONE
---

## [Editor Report · Acceptance letter]

13 Oct 2022

PONE-D-22-17677R1 

The Effect of a Telephone-based Intervention on Physical Activity after Stroke 

Dear Dr. Kim:

I'm pleased to inform you that your manuscript has been deemed suitable for publication in PLOS ONE. Congratulations! Your manuscript is now with our production department. 

Kind regards, 

on behalf of

Dr. Billy Morara Tsima 

Academic Editor

PLOS ONE